# High Fidelity Aggregated Planar Prior Assisted PatchMatch Multi-View Stereo

Submission Id: 1917

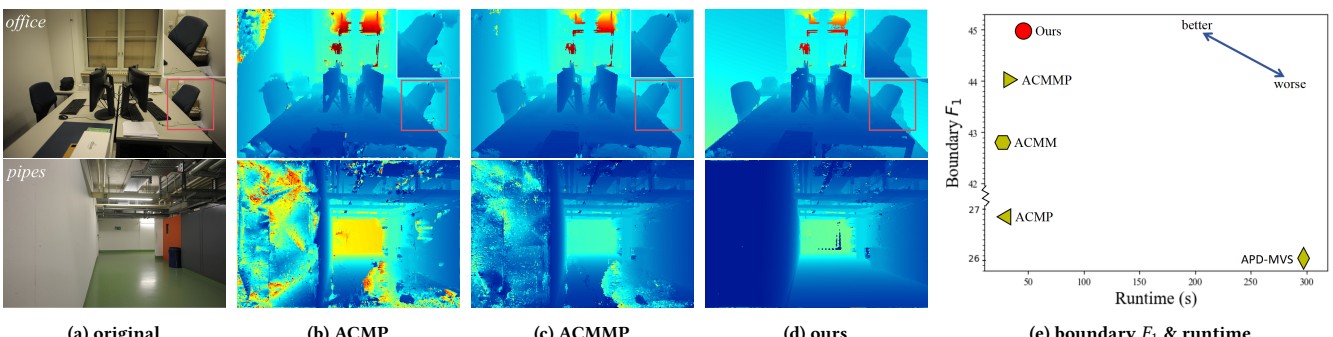

(a) original  (b) ACMP  (c) ACMMP  (d) ours  (e) boundary $F_1$ & runtime

**Figure 1: Qualitative results, boundary performance and runtime on ETH3D [30]. In *office* (a-d), with the help of boundary planes, we have more accurate (sharper) depth at the edge of the chair and reconstruct the stack of books behind the chair successfully. In *pipes* (a-d), we have more complete depth within large-scale weakly textured objects such as walls and floors due to object planes. (e) The y-axis represents $F_1$ score at the object boundary, and the x-axis represents the runtime. Compared to APD-MVS [39] with the best overall performance, we have a significant advantage both in boundary $F_1$ and runtime.**

## ABSTRACT

The quality of 3D models reconstructed by PatchMatch Multi-View Stereo remains a challenging problem due to unreliable photometric consistency in object boundaries and textureless areas. Since textureless areas usually exhibit strong planarity, previous methods used planar prior and significantly improved the reconstruction performance. However, their planar prior ignores the depth discontinuity at the object boundary, making the boundary inaccurate (not sharp). In addition, due to the unreliable planar models in large-scale low-textured objects, the reconstruction results are incomplete. To address the above issues, we introduce the segmentation generated from Segment Anything Model into PM pipelines for the first time. We use segmentation to determine whether the depth is continuous based on the characteristics of segmentation and depth sharing boundaries. Then we segment planes at object boundaries and enhance the consistency of planes in objects. Specifically, we construct **Boundary Plane** that fits the object boundary and **Object Plane** to increase consistency of planes in large-scale textureless objects. Finally, we use a probability graph model to calculate the **Aggregated Prior guided by Multiple Planes** and embed it into the matching cost. The experimental results indicate that our method

achieves state-of-the-art in terms of boundary sharpness on ETH3D. And it also significantly improves the completeness weakly textured objects. We also validated the generalization of our method on Tanks&Temples.

## CCS CONCEPTS

• **Computing methodologies** → **Reconstruction**; **Matching**.

## KEYWORDS

Multi-View Stereo, PatchMatch, Depth Estimation

**ACM Reference Format:**
Anonymous Author(s). 2018. High Fidelity Aggregated Planar Prior Assisted PatchMatch Multi-View Stereo. In *Proceedings of Make sure to enter the correct conference title from your rights confirmation emai (Conference acronym 'XX)*. ACM, New York, NY, USA, 10 pages. https://doi.org/XXXXXXX.XXXXXXX

## 1 INTRODUCTION

Enhancing the multimedia application's user experience requires the usage of interactive technologies. Virtual reality (VR) and augmented reality (AR), two cutting-edge interactive multimedia that offer interactive experiences in three dimensions, have garnered a lot of attention lately. A fundamental challenge in creating 3D content for VR and AR is to obtain 3D geometry through multi-view stereo. Numerous ideas originate from this line of thinking [27, 29, 41, 47, 52] and consistently raise the bar for reconstruction performances. These earlier techniques can be broadly categorized as traditional and deep learning-based methods.

Deep learning-based methods [20–22, 44, 47] have demonstrated remarkable power in MVS due to their exceptional capacity to extract robust visual features. In some datasets (e.g. DTU [1] and Tanks&Temples [15]), they achieve satisfactory results. However, most learning-based methods [20–22, 44] need to build the cost volume and increase the receptive field enormously when dealing with large-scale textureless regions, consequently leading to prohibitive memory consumption. Despite many efforts to reduce memory consumption [12, 24, 46, 48], learning-based methods still struggle to handle datasets with large-scale textureless areas or high-resolution images using mainstream GPU devices (fig. 7d). In addition, learning-based methods often rely on ground truth (GT), which is difficult to obtain on large-scale datasets. The above problems have led to poor performance of learning-based methods on the more challenging large-scale datasets (e.g. ETH3D [30]).

Many recently suggested traditional MVS methods [11, 29, 53] are essentially expanded versions of PatchMatch [4], which computes the matching cost based on a plane hypothesis between a fixed-size reference patch and patches in source images. PM methods require less memory since they find suitable matches by using a propagation and local refinement strategy, which avoids the construction of cost volume. PM methods have the advantages of low memory consumption and no need for ground truth, making it suitable for reconstruction tasks in large-scale scenes. Nevertheless, the matching cost will become unreliable when a patch is located in a textureless region because the receptive field lacks useful feature information [19]. [16, 42, 45] introduce a coarse fitting plane hypothesis based on the assumption that textureless regions often occur on flat surfaces (such as floors). ACMMP [42] uses Delaunay Triangulation of reliable pixels to construct planar models for textureless regions (fig. 2a), and embeds them as priors to the matching cost calculation. However, their boundaries are not sharp enough (*office* in fig. 1). That's because they don't fully consider the character of depth discontinuity at the object boundary, which is not in line with the plane assumption (green triangles in fig. 2c). On the other hand, ACMMP can lead to many errors in large-scale low-textured objects (*pipes* in fig. 1). This is due to the unreliability of pixel cost computations caused by photometric consistency failure.

To develop a memory friendly method that ensures sharp boundaries and completeness of weak texture objects, we combine segmentation [14] with the planar model of ACMMP. We notice the character that segmentation and depth share boundaries [54]. And many methods have also proven the mutually beneficial relationship between segmentation and depth recovery [7, 37]. We use segmentation to determine the continuity of depth, and then construct new planar models to improve the quality of planar priors.

Firstly, we use Delaunay triangulation to generate initial planar models (fig. 2a) and use SAM (Segment Anything Model [14]) to generate segmentation information. SAM will generate a mask for each object. We combine these masks as an identity map (**Id Map**, fig. 2b) where pixels belonging to the same object have the same Id value. Secondly, we detect erroneous planar models that span different objects (fig. 2c). Then we divide the error planar model into multiple small regions (collection of pixels with the same Id) and construct new planar models (**Boundary Plane**) for these small regions separately. In this way, planar models will fit the boundary of the object, making our depth map sharper. Thirdly, we

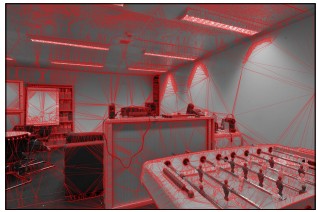
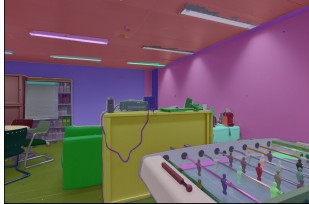

(a) Initial Plane      (b) Id Map

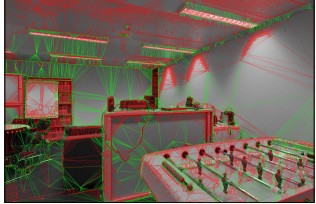
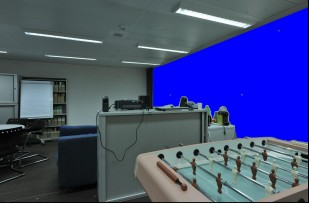

(c) Boundary Plane      (d) Object Plane

**Figure 2: (a) Initial planar models obtains from triangulation. Each triangle corresponds to a plane in 3D space. (b) Id Map obtained from SAM. Pixels belonging to the same object are labeled with the same color. (c) The green planar models span different objects. They will be divided into smaller regions to construct boundary planes. (section 3.2) separately. (d) Since planar models of the blue object are unreliable, we will construct a new object plane (section 3.3) for them.**

determine whether an object is reliable by using the texture and the size of the planar model in it. Then we attempt to fit an **Object Plane** using RANSAC for the object to enhance the consistency of these planar models (fig. 2d). Finally, we integrate the initial plane, boundary plane, and object plane as an **Aggregated Prior guided by Multiple Planes**, which is added to the matching cost.

In summary, we have the following contributions:

- Introducing SAM into the PatchMatch pipelines to construct boundary planes and object planes has improved the quality of the planar model.
- Embed aggregated prior guided by multiple planes into matching cost, making the depth sharper at object boundaries and more complete in textureless objects.
- Our method has the advantages of low memory consumption, strong generalization ability, and no need for GT, so it can be applied to large-scale scenes reconstruction.

## 2 RELATED WORK

### 2.1 PatchMatch-based MVS Methods

According to [31], MVS methods are divided into four categories: voxel-based methods [34], surface evolution [8] based methods, patch-based methods [10], and depth map based methods [4, 33]. The depth map based method has attracted much attention due to its advantages of easy transmission and parallelization.

Our method belongs to the last category, and we will only discuss this part of the content. PatchMatch multi-view stereo methods exploit the core idea of PatchMatch [2], sampling and propagation, to effectively estimate depth maps for each image. In recent

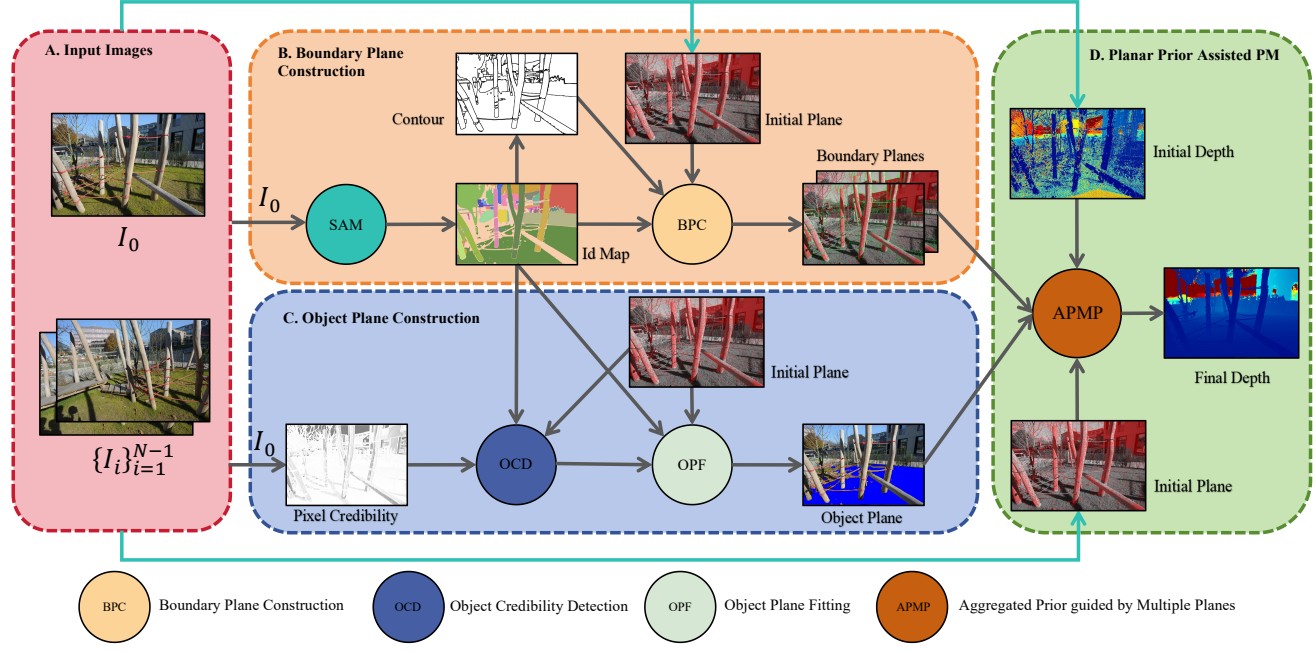

**Figure 3: Pipeline of HFP-MVS. Firstly, we can obtain the initial planar models and depth by running conventional PM and triangulation. Secondly, we combine the initial planar models with the segmentation extracted from SAM to generate Boundary Plane (section 3.2). Thirdly, we determine whether an object is credible, and try to construct an Object Plane (section 3.3) for the incredible object. Finally, we integrate the initial plane, boundary plane, and object plane to generate an Aggregated Prior guided by Multiple planes (section 3.4) for each pixel, which will be embedded to the matching cost.**

years, [11, 29] have significantly improved the effectiveness of PM-based methods. ACMH designs an adaptive checkerboard sampling strategy to propagate more reliable hypotheses. ACMM [43] incorporates pyramid structure and geometric consistency into PM, allowing it to effectively analyze rich textured regions and small textureless regions but still being unable to estimate the depth for large-scale textureless regions very well. On the basis of ACMM, APD-MVS [39] adaptively deform the patch of unreliable pixel to extend the receptive field until it covers enough reliable pixels. APD-MVS greatly improves the completeness of depth maps, but its performance at boundaries is poor. Differently, [16, 28] divide images into superpixels and fit a plane for each superpixel. ACMP [45] uses the Delaunay Triangulation of reliable pixels to propose a coarse planar model for large-scale textureless regions. It adds the planar model as a prior to patch matching. ACMP significantly improves completeness in textureless regions. ACMMP [42] integrates ACMP and ACMM, resulting in improved performance. However, ACMP and ACMMP don't consider that the depth near the object boundary is often discontinuous, and the coarse plane in such a depth discontinuous area will provide an incorrect prior. Moreover, due to the unreliable matching cost calculation and difficulty in finding reliable pixels in large-scale textureless objects, the planes on these objects are often wrong.

## 2.2 Learning-based MVS Methods

Learning-based techniques have become popular when [47] used deep learning for depth map estimation. Many works are dedicated to reducing the heavy calculation of cost volume with a coarse-to-fine strategy [12, 24, 46, 49] and RNNs [48]. Meanwhile, some researches formulate a more reliable cost volume, such as the visibility of ViS-MVSNet [51]. To achieve a more robust feature extraction, AA-RMVSNet [40] proposes an adaptive aggregation module implemented using deformable convolution. [9, 38] introduce the transformer structure into the MVS task to obtain global feature information. Transformer-based [6, 9, 18, 23, 38] introduce finely designed external structures for feature extraction but do not completely use the geometric clues embedded in the MVS scenarios [52]. Learning-based methods use networks to learn features, resulting in performance surpassing traditional methods on some datasets [1, 15]. However, in large-scale scene datasets [30], there are problems such as the inability to obtain accurate and complete ground truth, as well as huge memory consumption, which leads to poor deep learning methods. Traditional methods still are SOTA on these datasets due to their advantages of low graphics memory consumption and strong generalization ability.

There are also unsupervised methods such as [3, 13, 17, 25, 26], which have made significant progress. However, they only use images as supervised signals, which leads to their inability to generate correct geometric information.

# 3 METHOD

Given a set of images $\{I_i\}_{i=1}^M$ and the corresponding camera parameters $\{P_i\}_{i=1}^M$, our algorithm estimates the depth map of each image. For reference image $I_0$ with its source images $\{I_i\}_{i=1}^{N-1}$, We first use the conventional PM to obtain the initial depth map, and based on this, we use Delaunay Triangulation to obtain the initial planar models (section 3.1). Secondly, with the assistance of segmentation information from SAM (section 3.2), generate boundary planar models (section 3.2) and object planar models (section 3.3). Thidly, we generate an aggregated prior guided by multiple planaes and add it into matching cost (section 3.4). Finally, we obtained the depth map by iterating the PM process.

## 3.1 Preliminary

*3.1.1 PatchMatch.* Firstly, PM performs random initialization, generating depth and normal vectors randomly for each pixel in $I_0$ as the initial planar hypothesis. The second step is propagation. For pixel $p$, PM use the plane hypotheses of eight pixels in its neighborhood as the hypothetical space. Thirdly, calculate the matching cost for each planar hypothesis in the hypothetical space. The reference patch is represented by a square window $B_p$ that is centered on $p$. PM assumes that pixels in $B_p$ are all on the same plane. Given a planar hypothesis $\theta_i = [n_i^\top, d_i]$, the projected patch $B_p^j$ on the source image $I_j$ for $B_p$ can be obtained by homography [32] formula. PM obtains the dissimilarity score between $B_p$ and $B_p^j$ through one minus the NCC score [42]. After aggregating each dissimilarity score using viewpoint weights [42], PM obtains the final matching cost $c_{photo}(\theta_i)$. For each planar hypothesis in the hypothetical space, we use the above process to obtain the corresponding cost. The fourth step is refinement, where PM selects the planar hypothesis with the minimum cost and optimizes it by introducing perturbations and randomness. PM use the optimized hypothesis as the new planar hypothesis for $p$. By iterating through the last three steps, PM can acquire a final depth map. Matching cost calculation is a key step in pipelines as it is crucial for the accuracy of depth maps.

*3.1.2 Planar Prior.* Here we review the planar prior in our baseline ACMMP. The planar prior is divided into two steps. Firstly, ACMMP constructs the planar model. Secondly, ACMMP calculates the planar prior and adds it to the matching cost calculation of PM.

Specifically, ACMMP uses above PM process to generate an initial depth and corresponding cost map. Then, it select the pixels with the lowest local cost as the reliable points. It takes these points as vertices and uses Delaunay Triangulation to obtain triangles where adjacent triangles share the vertices (fig. 2a). Each triangle represents a planar model in the reference camera coordinate system. The normal vector of a planar model and the distance from the origin of the camera coordinate system can be calculated based on the camera coordinates of the three triangle vertices. ACMMP assumes that the pixels inside the triangle are all located on the corresponding planar model. It calculates the depth and normal vector $\theta_p = [n_p^\top, d_p]$ for each pixel based on the planar model. ACMMP calculates a planar prior, i.e. a prior probability, for each pixel by calculating the difference between $\theta_p$ and its planar hypothesis $\theta_i$

$$P(\theta_i|\theta_p) = \gamma + e^{-\frac{(d_i-d_p)^2}{2\lambda_d}} \cdot e^{-\frac{arccos^2 n_i^\top n_p}{2\lambda_n}}, \quad (1)$$

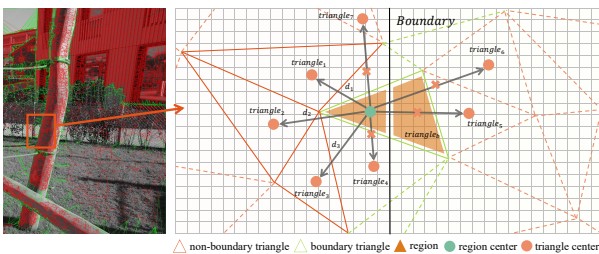

Figure 4: Boundary plane construction. We divide $triangle_b$ into two regions using Id. For a region, we use nearby non-boundary triangles to provide boundary planes. $triange_4$ will be excluded because it also spans different objects. $triange_{5,6}$ will be excluded because $Id(triangle)$ is different from $Id(region)$. $triange_7$ will be excluded because the distance is too far. Finally, we will select $triangle_{1,2,3}$ as boundary planes of the region.

where $\lambda_d$ is the bandwidth of depth difference, $\lambda_n$ is the bandwidth of normal difference. The higher the probability value, the closer the $\theta_i$ is to $\theta_p$, and the more credible the $\theta_i$ is. Next, ACMMP incorporates planar priors into the matching cost calculation

$$c_{p-photo}(\theta_i) = \frac{c_{photo}(\theta_i)^2}{\alpha} - log[P(\theta_i|\theta_p)]. \quad (2)$$

When the photometric consistency cannot reflect hypothesis changes in textureless areas, the planar prior will play a major role in the hypothesis updating. $\alpha$ is a empirical constant [42].

## 3.2 Boundary Plane Construction

We input $I_0$ into SAM, which will generate masks for objects. After performing merging of these masks, we will get an identity map (Id Map) where pixels belonging to the same object will have the same Id value, as shown in fig. 2b. Then we use Canny [5] operator to extract initial contours on Id Map. For robustness, we will take a circular neighborhood with 11 centered around the initial contour pixel and label all pixels as contour pixels.

We use the planar model constructed using the method in section 3.1 as the initial planar model. However, we can notice that many planes crossing the boundaries of objects (green triangles in fig. 2c), which may construct many wrong oblique planes with part points in the foreground and part points in the background. These planes will cause errors in the cost calculation of the boundary area. To this end, we will detect these erroneous planar models and repair them using the correct planar models from their neighborhoods.

First, we use contour to determine whether the triangle crosses the boundary. If the number of contour pixels in a triangle exceeds the threshold $\alpha_{nb}$, it is determined to be a boundary triangle. Otherwise it will be judged as a non-boundary triangle. For a non-boundary triangle $t$ (red triangles in fig. 2c), we regard it as a correct initial planar model and will use it to fix the boundary triangle. We use the center of gravity $[\frac{x_1+x_2+x_3}{3}, \frac{y_1+y_2+y_3}{3}]$ to represent its position $pos_t$. And calculate the pixels' Id distribution within the $t$, taking the Id with the highest number of pixels as $id_t$.

For a boundary triangle, we divide it into multiple regions (fig. 4). A region is a collection of pixels with the same Id. Here lies a natural

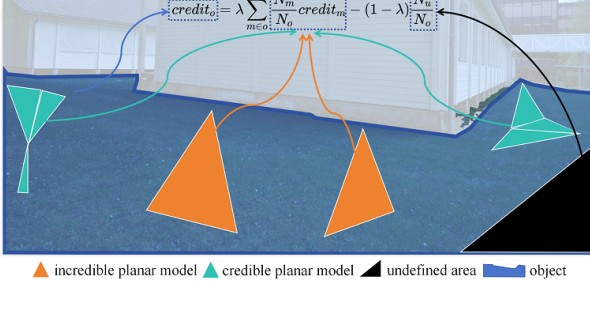

▲ incredible planar model   ▲ credible planar model   ▲ undefined area   ■ object

**Figure 5: Object Plane Construction. Firstly, we evaluate the credibility of the planar model based on pixel credibility and model size. Cyan triangles and orange triangles represent planar models with high and low credibility, respectively, and the black area represents that there is no planar model here. Then we aggregate the credibility to obtain the credibility of the blue object. For an incredible object, we sample low-cost points in planar models with high credibility (Cyan triangles) and run RANSAC to fit an object plane for it.**

assumption: we believe that the depth changes of pixels belonging to the same object are often continuous, while the depth changes of pixels belonging to different objects are often discontinuous. So we use nearby non-boundary triangles to help construct planar models for these regions. For a region $r$, we use the average coordinates of all pixels in it as the location of the region $pos_r$. Next, we calculate the distance between the region and the non-boundary triangle belonging to the same object. We add the triangles with distance calculated from eq. (3) less than $\varepsilon$ to the set $neighbor_r$.

$$d = \|pos_r - pos_t\|_2 . \tag{3}$$

For the pixel $p$ in $r$, its possible planar models are $neighbor_r$. We will calculate a weight for each plane in $neighbor_r$ that takes into account distance and matching costs. For $plane_j$ in $neighbor_r$, we define its weight as

$$w_{bj} = e^{-\frac{c_{photo}^2(\theta_{bj})}{\beta}} \cdot e^{-\frac{d^2}{\gamma}}, \tag{4}$$

where $\theta_{bj}$ represents the planar hypothesis when $p$ is on the $plane_j$, and $d$ represents the distance calculated by eq. (3). The negative exponential form ensures that the output value is within the range of 0 to 1. $\beta$ is a small constant that makes it convenient to set the weight of a planar model with cost exceeding the threshold to 0. This can mitigate the impact of outliers on weight calculation. And because the difference between distances is often much greater than the difference between costs, we use a large constant $\gamma$ to control the impact of distance on weights. The closer the distance and the lower the matching cost, the more likely the pixel $p$ is to be on the $plane_j$, and thus the weight of $plane_j$ will be greater.

We construct multiple possible planar models $neighbor_r$ for a region and calculate weights $w_b$ for each planar model. They will be applied in section 3.4 to construct our aggregated prior guided by multiple planes.

## 3.3 Object Plane Construction

In addition to the errors in depth discontinuous areas, we found that on large-scale textureless objects, the quality of the planar model is also poor. That's because severe low texture can lead to unreliable depth calculation, and also make the selection of reliable points difficult. Ultimately, it leads to a decrease in the quality of the planar model. Therefore, we detect the credibility of object reconstruction results and attempt to construct an object plane to improve the quality of unreliable plane models.

*3.3.1 Object Credibility Detection.* We notice that objects are composed of multiple planar models, which in turn are composed of multiple pixels. Therefore, we define the credibility of pixels, planar models, and objects in ascending order.

According to [28], we define the credibility of a pixel $p$ as

$$credit_p = \frac{Var + \epsilon_{var}}{Var + \frac{\epsilon_{var}}{t_{min}}}, \tag{5}$$

where $Var$ is the variance of the 5×5 patch around the pixel. A larger variance means that the features around the pixel are richer, and the probability of calculating the correct depth through matching cost is also higher, so it is considered that the credibility of the pixel is higher. $\epsilon_{var}$ and $t_{min}$ are two empirical values, and the same values as [28] are used here.

For a planar model $m$ in object $o$, we define its credibility as

$$credit_m = \frac{1}{N_m} \sum_{p \in m} credit_p \cdot e^{-\frac{(N_m/N_o)^2}{\eta}}, \tag{6}$$

where $N_m$ and $N_o$ represent the number of pixels in $m$ and $o$, respectively. The first term represents the average credibility of pixels in $m$. The greater the first term, the higher the credibility of $m$. This is because the construction of a planar model depends on the initial depth and cost generated by the conventional PM process. The stronger the texture is, the more reliable the matching cost calculation of PM will be, making the planar model more reliable. The $N_m/N_o$ in the second item represents the proportion of the plane model to the object. The larger the $N_m/N_o$ is, the lower credibility of $m$ will be. That's because, compared to small planar models, large planar models mean difficulty in selecting reliable points.

Finally, by aggregating the credibility of planar models, we can obtain the credibility of object $o$ as

$$credit_o = \lambda \sum_{m \in o} \frac{N_m}{N_o} credit_m - (1 - \lambda) \frac{N_u}{N_o}, \tag{7}$$

where $N_u/N_o$ represents the proportion of the area without a planar model to the size of the object. We use $\lambda$ to control the weights of the two terms. There are many areas at the boundaries of the image that do not have a planar model, so when calculating the credibility of the $o$, this also needs to be taken into account. We mark objects with $credit_o$ below the threshold $\tau$ as unreliable objects.

*3.3.2 Object Plane Fitting.* For an unreliable object $o$, we have a pixel set $R = \{p | p \in m, m \in o, credit_p > \xi, credit_m > \pi, cost_p < \rho\}$. This is equivalent to selecting the low-cost credible points from the credible planar model, and we run RANSAC algorithm on $R$.

In each iteration, we randomly select three pixels from $R$ and construct a plane with their world coordinates, and calculate the interior point rate of $R$ on this plane. We choose the plane with the

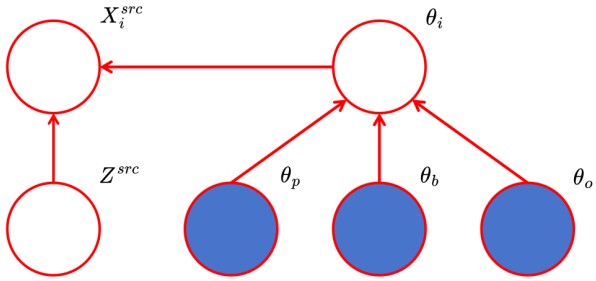

**Figure 6: Graphical model of planar prior assistance. Given multiple planar models ($\theta_p$, $\theta_b$, $\theta_o$), the observation $X_i^{src}$ on source images, and the visibility information $Z^{src}$, the optimal hypothesis $\theta^*$ is inferred.**

highest interior point rate as the final plane. If the interior point rate of the final fitting plane is more than $\varphi$, it indicates that the object has successfully constructed an object plane. Otherwise, we believe that the object is not a plane.

If an object plane is successfully constructed, each pixel in the $o$ will have an object planar model $plane_o$. And we calculate a weight for $plane_o$

$$w_o = \gamma_o + e^{-\frac{c_{photo}^2(\theta_o)}{\alpha}}, \tag{8}$$

where $\theta_o$ represents the planar hypothesis when $p$ is on the $plane_o$, and $\gamma_o$ is a constant used to align with the weight of other planar models (boundary or initial planar models). $w_o$ will be applied in section 3.4 to construct aggregated prior guided by multiple planes.

## 3.4 Planar Prior Assistance

So far, we have three types of planar models, namely the initial planar model, the boundary planar model that enhances boundary sharpness, and the object planar model that enhances the completeness of weakly textured objects. We will generate an aggregated prior guided by multiple planes and embed it into the calculation of matching cost in this section.

Our innovative matching cost assisted with aggregated prior guided by multiple planes is derived through a probabilistic graphical model. To construct the graphical model, we define the patch on pixel $p$ as $X^{ref}$. Also, the patches observed on all source images via $\theta_i$ are $X_i^{src}$, and the visibility information of all source images is assumed to be $Z^{src}$. $\theta_p$ represents the initial plane, $\theta_b$ represents the boundary planes (there may be more than one boundary plane) and $\theta_o$ represents object plane. A pixel may have one or two types of planar models, represented by $\Theta$, which will be reflected in eq. (13).

The fig. 6 shows the graphical model of our approach. The joint probability is

$$P(\theta_i, X_i^{src}, Z^{src}, \Theta) \propto P(X_i^{src}|\theta_i, Z^{src})P(\theta_i|\Theta). \tag{9}$$

In this way, the maximum a posteriori estimate of the plane hypothesis $\theta^*$ is given by

$$\theta^* = argmax P(\theta_i|X_i^{src}, Z^{src}, \Theta). \tag{10}$$

The above posterior can be factorized as

$$P(\theta_i|X_i^{src}, Z^{src}, \Theta) \propto P(X_i^{src}|\theta_i, Z^{src})P(\theta_i|\Theta). \tag{11}$$

Next, the likelihood function can be defined as follows,

$$P(X_i^{src}|\theta_i, Z^{src}) = e^{-\frac{c_{photo}(\theta_i)^2}{\alpha}}. \tag{12}$$

This function encodes the photometric consistency, making the low multi-view aggregated photometric consistency cost have a high probability. To solve errors in object boundaries and large-scale low-textured objects, we define different planar prior based on whether three types of planar models are constructed as follow

$$P(\theta_i|\Theta) = \begin{cases} P(\theta_i|\theta_p) & \theta_p \\ \frac{\sum_j w_{bj}P(\theta_i|\theta_b)}{\sum_j w_{bj}} & \theta_b \\ \frac{\sum_j w_{bj}P(\theta_i|\theta_p)+w_oP(\theta_i|\theta_o)}{\sum_j w_{bj}+w_o} & \theta_b, \theta_o \\ \frac{w_pP(\theta_i|\theta_p)+w_oP(\theta_i|\theta_o)}{w_p+w_o} & \theta_p, \theta_o \end{cases}. \tag{13}$$

Each row of formulas corresponds to a situation of constructing planar models. The first and second rows indicate that only the initial plane or boundary planes are provided respectively, the third row indicates that both the boundary plane and object plane are provided, and the fourth row indicates that both the initial plane and object plane are provided. $P(\theta_i|\theta_p)$, $P(\theta_i|\theta_b)$, and $P(\theta_i|\theta_o)$ can be calculated by eq. (1). $w_{bj}$ and $w_o$ represent the weight of boundary planar model $\theta_{bj}$ and object planar prior $\theta_o$, respectively. They can be calculated in section 3.2 and section 3.3. $w_p$ is a constant used to represent the weight of the initial planar model. Finally, we substitute eq. (11), eq. (12), and eq. (13) into eq. (10) and take the negative logarithm algorithm to get the following aggregated prior guided by multiple planes assisted matching cost

$$c_{mp\_photo} = \frac{c_{photo}(\theta_i)^2}{\alpha} - logP(\theta_i|\Theta). \tag{14}$$

Note that, the first term that encodes photometric consistency is the main component in the above equation. This indicates that in well-textured areas, the photometric consistency will change more obviously than the planar prior [45]. When the photometric consistency cannot reflect hypothesis changes in low-textured areas, thanks to the aggregated prior, our cost calculation can be more accurate at object boundaries and within low-textured objects.

We also use the same pyramid structure as ACMMP. After iterating through the process of PM, we can obtain the depth.

## 4 EXPERIMENTS

### 4.1 Datasets and Results

To verify the effectiveness of our method in large-scale scenarios, we use high-resolution images in ETH3D [30] and Tanks&Temples [15] in our experiments. ETH3D is a more challenging benchmark. Firstly, it has a more diverse range of scene types (from man-made indoor and outdoor scenes to natural scenes containing a large amount of vegetation), which requires more generalized methods to avoid overfitting. Secondly, the complexity of the scene and the drastic changes in viewpoints result in a large number of weak textures, occlusion, and other difficult reconstruction areas. Finally, extremely high image resolution (6,048 × 4,032) also requires memory and computation efficient methods. Learning-based methods often fail in such challenges, however, due to the low memory consumption and excellent generalization ability of PM-based methods,

**Table 1: Quantitative results on ETH3D benchmark. We have achieved state-of-the-art overall performance. Best results are in bold, while second-best results are underlined. ACMMP is our baseline.**

| Method | Test | | | | | | Train | | | | | |
|---|---|---|---|---|---|---|---|---|---|---|---|---|
| | 2cm | | | 10cm | | | 2cm | | | 10cm | | |
| | $F_1$ | Comp | Acc | $F_1$ | Comp | Acc | $F_1$ | Comp | Acc | $F_1$ | Comp | Acc |
| PatchmatchNet [36] | 73.12 | 77.46 | 69.71 | 91.91 | 92.05 | 91.98 | 64.21 | 65.43 | 64.81 | 85.70 | 83.28 | 89.98 |
| GBi-Net [24] | 78.40 | 75.65 | 82.02 | 91.35 | 86.67 | 96.99 | 70.78 | 69.21 | 73.17 | 90.21 | 86.16 | 95.21 |
| IterMVS-LS [35] | 80.06 | 76.49 | 84.73 | 92.29 | 88.34 | 96.92 | 71.69 | 66.08 | 79.79 | 88.60 | 82.62 | 96.35 |
| MVSTER [38] | 79.01 | 82.47 | 77.09 | 93.20 | 92.71 | 94.21 | 72.06 | 76.92 | 68.08 | 91.73 | 91.91 | 91.97 |
| EPP-MVSNet [23] | 83.40 | 81.79 | 85.47 | 95.22 | 93.75 | 96.84 | 74.00 | 67.58 | 82.76 | 92.13 | 87.72 | 97.29 |
| COLMAP [29] | 73.01 | 62.98 | **91.97** | 90.40 | 84.54 | **98.25** | 67.66 | 55.13 | **91.85** | 87.61 | 79.47 | **98.75** |
| TAPA-MVS [28] | 79.15 | 74.94 | 85.71 | 92.30 | 90.35 | 94.93 | 77.69 | 71.45 | 85.88 | 93.69 | 90.98 | 96.79 |
| ACMM [43] | 80.78 | 74.34 | 90.65 | 92.96 | 88.77 | 98.05 | 78.86 | 70.42 | 90.67 | 91.70 | 86.40 | 98.12 |
| ACMP [45] | 81.51 | 75.58 | 90.54 | 92.62 | 88.71 | 97.47 | 79.79 | 72.15 | 90.12 | 92.03 | 87.15 | 97.97 |
| APD-MVS [39] | 87.44 | **85.93** | 89.54 | 96.95 | **96.95** | 97.00 | **86.84** | **84.83** | 89.14 | 97.12 | **96.79** | 97.47 |
| ACMMP [42] | 85.89 | 81.49 | 91.91 | 96.27 | 94.67 | 98.05 | 83.42 | 77.61 | 90.63 | 95.54 | 93.32 | 97.99 |
| HFP-MVS (ours) | **87.58** | 84.73 | 91.30 | **97.14** | 96.42 | 97.94 | 86.15 | 82.48 | 90.32 | **97.14** | 96.10 | 98.22 |

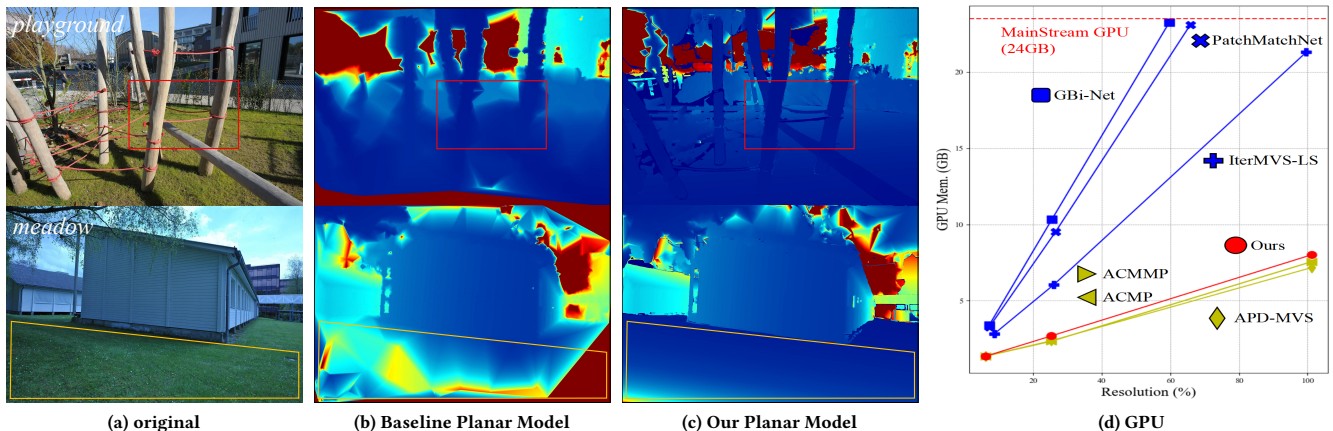

**(a) original**  **(b) Baseline Planar Model**  **(c) Our Planar Model**  **(d) GPU**

**Figure 7: The planar model and graphics memory consumption on ETH3D. In *playground* (a-c), due to the lack of consideration for depth discontinuity, the baseline constructed many error planar models that cross tree boundaries, while our planar model fits well with tree boundaries. In *meadow* (a-c), due to the failure of matching on large-scale weakly textured objects, most of the planar models constructed by the baseline on grassland are incorrect, while the object plane we constructed can improve the quality of the planar model on grassland. (d) Compared to learning-based methods (blue), PM-based methods (red and yellow) has significantly lower memory consumption, making it more suitable for large-scale scenes.**

the SOTA under the ETH3D benchmark are still PM-based methods. Tanks&Temples dataset also contains large-scale scenes but has a smaller resolution (about 1,920 × 1,080). We use it to demonstrate the generalization ability of our method. All our experiments are performed on a single NVIDIA Tesla V100 GPU.

We use fusion technique like [42] to obtain point clouds on ETH3D dataset. fig. 1 shows a qualitative comparison, and it is evident that our method takes into account both the sharpness of object boundaries and the completeness of textureless objects. The quantitative analysis is displayed in table 1 with the first group being based on learning and the second being traditional. Furthermore, only a few learning-based approaches, such [24, 36, 51], can successfully complete the reconstruction task on this dataset due

to the high resolution of the images (fig. 7d). However, their performance is still insufficient. We also validated the generalization of our method on Tanks&Temples. It can be seen that our method significantly improves the performance of the baseline.

## 4.2 Memory Comparison

For all methods, we set the number of source pictures to 10 and the image size to 6, 221 × 4, 146 (ETH3D) at 100% resolution (8.04% corresponds to Tanks&Temples) in order to compare memory costs. As shown in fig. 7d, compared to the learning-based method (blue), PM methods (yellow and red) has significantly less memory consumption. Compared to the baseline, our method hardly increases

**Table 2: Quantitative results on Tanks&Temples benchmark. Compared to the baseline (ACMMP [42]), our method has achieved significant improvement. Best results are in bold, while second-best results are underlined.**

| Method | Intermediate | | | | | | | | | Advanced | | | | | | |
|---|---|---|---|---|---|---|---|---|---|---|---|---|---|---|---|---|
| | Mean | Fam. | Fra. | Hor. | Lig. | M60. | Pan. | Pla. | Tra. | Mean | Aud. | Bal. | Cou. | Mus. | Pal. | Tem. |
| PatchmatchNet [36] | 53.15 | 66.99 | 52.64 | 43.24 | 54.87 | 52.87 | 49.54 | 54.21 | 50.81 | 32.31 | 23.69 | 37.73 | 30.04 | 41.80 | 28.31 | 32.29 |
| CasMVSNet [12] | 56.84 | 76.37 | 58.45 | 46.26 | 55.81 | 56.11 | 54.06 | 58.18 | 49.51 | 31.12 | 19.81 | 38.46 | 29.10 | 43.87 | 27.36 | 28.11 |
| ElasticMVS [50] | 57.88 | 69.11 | 63.74 | 43.43 | 62.61 | 59.41 | 51.85 | 59.35 | 53.48 | 37.81 | 21.35 | 42.96 | 38.30 | 54.03 | 31.71 | 38.55 |
| VisMVSNet [51] | 60.03 | 77.40 | 60.23 | 47.07 | 63.44 | 62.21 | 57.28 | 60.54 | 52.07 | 33.78 | 20.79 | 38.77 | 32.45 | 44.20 | 28.73 | 37.70 |
| COLMAP [29] | 42.14 | 50.41 | 22.25 | 25.63 | 56.43 | 44.83 | 46.97 | 48.53 | 42.04 | 27.24 | 16.02 | 25.23 | 34.70 | 41.51 | 18.05 | 27.94 |
| PCF-MVS [16] | 55.88 | 70.99 | 49.60 | 40.34 | 63.44 | 57.79 | 58.91 | 56.59 | 49.40 | 35.69 | 28.33 | 38.64 | 35.95 | 48.36 | 26.17 | 36.69 |
| ACMM [43] | 57.27 | 69.24 | 51.45 | 46.97 | 63.20 | 55.07 | 57.64 | 60.08 | 54.48 | 34.02 | 23.41 | 32.91 | 41.17 | 48.13 | 23.87 | 34.60 |
| ACMP [45] | 58.41 | 70.30 | 54.06 | 54.11 | 61.65 | 54.16 | 57.60 | 58.12 | 57.25 | 37.44 | 30.12 | 34.68 | 44.58 | 50.64 | 27.20 | 37.43 |
| ACMMP [42] | 59.38 | 70.93 | 55.39 | 51.80 | 63.83 | 55.94 | 59.47 | 59.51 | 58.20 | 37.84 | 30.05 | 35.36 | 44.51 | 50.95 | 27.43 | 38.73 |
| HFP-MVS (ours) | 61.00 | 73.68 | 57.96 | 51.13 | 65.44 | 61.53 | 61.36 | 61.15 | 55.79 | 39.81 | 29.91 | 45.53 | 39.74 | 52.22 | 28.91 | 42.54 |

**Table 3: $F_1$ at object boundaries (ETH3D training set). We have state-of-the-art performance near object boundaries.**

| Method | 2cm | | | 10cm | | |
|---|---|---|---|---|---|---|
| | $F_1$ | Comp | ACC | $F_1$ | Comp | ACC |
| APD-MVS [39] | 25.80 | 15.92 | 84.16 | 51.23 | 37.36 | 94.24 |
| ACMP [45] | 26.87 | 16.44 | 86.88 | 49.51 | 34.82 | 96.56 |
| ACMM [43] | 42.80 | 28.49 | 91.88 | 63.65 | 48.47 | 98.52 |
| ACMMP [42] | 44.03 | 29.53 | 92.57 | 66.51 | 51.92 | 98.55 |
| ours | 44.98 | 30.37 | 93.08 | 67.65 | 53.28 | 98.83 |

memory consumption as we only store the Id Map and the new planar models separately.

## 4.3 Boundary Performance

From *playground* in fig. 7, the baseline constructs many erroneous planar models near the boundaries of trees, while our planar models are more closely aligned with the boundaries of trees. The quality of the planar model will affect the quality of the depth map. From *office* in fig. 1, it can be seen that our depth map has significantly sharper boundaries and successfully reconstructed the stack of books behind the chair, which also indicates that our method has better reconstruction ability in complex scenes. We also conducted a quantitative analysis of the boundaries. We only fuse the depth information located at the object boundary (the contour in section 3.2) on ETH3D. As shown in table 3, our method has best $F_1$ scores at the object boundary compared to other methods.

## 4.4 Ablation Study

Our method consists of two modules, namely boundary planar prior (BP) and object planar prior (OP). From *playground* in fig. 7, we can see that the boundary plane is more closely aligned with the object boundary. From *meadow* in fig. 7, it can be seen that there are a large number of errors in the initial planar model on large-scale weakly textured objects such as grasslands. Our method can detect these low credibility objects and fit them to an object plane. We also conducted quantitative analysis on ETH3D. In table 4, boundary planar prior can improve both completeness and accuracy, while object planar prior mainly enhances completeness. When we combine two modules, we can achieve the best performance.

**Table 4: Ablation experiment on ETH3D training set. ours/OP and ours/BP represent the method after removing object planar prior and boundary planar prior, respectively.**

| Method | 2cm | | | 10cm | | |
|---|---|---|---|---|---|---|
| | $F_1$ | Comp | Acc | $F_1$ | Comp | Acc |
| baseline [42] | 83.42 | 77.61 | 90.63 | 95.54 | 93.32 | 97.99 |
| ours/OP | 85.49 | 80.61 | 91.26 | 96.91 | 95.58 | 98.30 |
| ours/BP | 84.73 | 80.86 | 89.28 | 96.34 | 95.13 | 97.67 |
| ours | 86.15 | 82.48 | 90.34 | 98.14 | 96.10 | 98.22 |

**Table 5: $F_1$ in textureless objects (ETH3D training set).**

| Method | 2cm | | | 10cm | | |
|---|---|---|---|---|---|---|
| | $F_1$ | Comp | ACC | $F_1$ | Comp | ACC |
| ACMP [45] | 63.76 | 50.07 | 90.77 | 76.38 | 63.29 | 98.30 |
| ACMMP [42] | 68.01 | 54.75 | 91.54 | 81.04 | 69.63 | 98.36 |
| ours | 70.78 | 58.81 | 90.5 | 82.31 | 71.45 | 98.44 |

To demonstrate the performance of our algorithm at the textureless object, we only fuse the depths located at these objects (The object judged to be incredible and successfully constructed Object Plane for it). We can see quantitative and qualitative experimental results in table 5 and *pipes* in fig. 1, respectively.

## 5 CONCLUSION

We present aggregated prior guided by multiple planes in this study and implement PM-based MVS technique, HFP-MVS. We introduce segmentation of SAM into the PM pipelines for the first time. We greatly enhance the accuracy of depth at object boundaries and the completeness of depth on large-scale weakly textured objects. Meanwhile, our method has the advantages of low graphics memory consumption and strong generalization, making it more suitable for reconstructing large-scale scenes (such as ETH3D). In addition, our method has strong interpretability and can serve as a prior for learning-based methods to improve their performance. However, for non-planar large-scale low-textured objects, our algorithm's planar prior quality remains unassured. Future research might address this issue by using a curved surface prior.

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
