# OpenReview forum: "High Fidelity Aggregated Planar Prior Assisted PatchMatch Multi-View Stereo"
_acmmm.org/ACMMM/2024/Conference — MM2024 Poster_

### Official Review · Reviewer_k2qS · 2024-05-22

**Rating:** 2
**Confidence:** 3

**Summary:**

This proposed model HFP-MVS introduces Segment Anything Model (SAM) into the PatchMatch pipelines to construct boundary planes and object planes has improved the quality of the planar model. It constructs boundary plane that fits the object boundary and object plane to increase consistency of planes in large-scale textureless objects. Then, it uses a probability graph model to calculate the aggregated prior guided by multiple planes and embed it into the matching cost. Extensive experimental results demonstrate the effectiveness of the proposed model HFP-MVS.

**Strengths:**

1. Introducing SAM into the PatchMatch pipelines to construct boundary planes and object planes has improved the quality of the planar model.
2. Extensive experimental comparisons and ablations are conducted on various widely-used benchmark datasets, which demonstrate that the proposed model HFP-MVS consistently outperforms state-of-the-art methods.

**Limitations:**

1. This paper uses a large-scale Segment Anything Model (SAM) with powerful capacities of generalization and discrimination to generate segmentation information for reconstruction performance. Compared with other segmentation models, SAM can provide more accurate segmentation information, which naturally leads to better results for the task of multi-view stereo. But in the section of ablation study, this paper lacks a direct comparison over the impact of different segmentation models for the final results. Thus, it is difficult to identify the source of the performance improvement.
2. This paper does not provide sufficient explanation and discussion on experimental settings. For example, in the section 4.1, there is a sentence "All our experiments are performed on a single NVIDIA Tesla V100 GPU" in this paper.
3. In the Main Result, only one work from 2023 is chosen for quantitative analysis. It should be appropriate to add recent works.
4. Some bugs. Some undefined abbreviations exist in the sections of Abstract and Introduction, such as “PM” and “MVS”. In section 1, the clear definition of the MVS task is not given. This could lead to ambiguity for readers trying to understand the incremental benefits of the proposed approach HFP-MVS over existing MVS methods.

**Suitability:**

2

---

### Official Review · Reviewer_6E3y · 2024-05-30

**Rating:** 4
**Confidence:** 4

**Summary:**

This paper improves ACMMP by introducing the results of the Segment Anything Model (SAM) to correct and optimize the planar prior generated in ACMMP: 1) planes spanning different objects are split, 2) planes belonging to the same object are reaggregated. The corrected planar prior then guides the MVS to achieve better reconstruction results.

**Strengths:**

1. By applying the results of SAM to obtain the edges of the objects and splitting the planar prior that spans multiple objects, the incorrect planar priors that can cause negative effects to MVS will be reduced, thus improving the accuracy of the reconstruction.
2. By aggregating planar priors, this method can better handle large-scale textureless objects.
3. This method does not introduce much more additional runtime and memory consumption compared to ACMMP, but achieves better results than ACMMP.

**Limitations:**

1. There is no special treatment for non-planar areas. If some textureless areas are non-planar and they cannot be separated using SAM, such aggregation may lead to worse results.
2. To demonstrate the improvement at the boundary areas, it is better to give a visual result of the point cloud, which is more intuitive.
3. There is no detailed explanation of how the boundary score is calculated. From my understanding, the edges are all strong textures. If only the depths on the edges are fused, the scores of these methods will not be so different unless the depths near the edges are also fused.

**Suitability:**

2

---

### Official Review · Reviewer_xuRR · 2024-06-01

**Rating:** 4
**Confidence:** 2

**Summary:**

This paper focuses on improving multi-view stereo (MVS) reconstruction by addressing challenges in object boundaries and textureless areas. The paper presents a SAM-based framework, High Fidelity Aggregated Planar Prior Assisted PatchMatch Multi-View Stereo (HFP-MVS), which introduces segmentation from the Segment Anything Model (SAM) into PatchMatch pipelines. The paper proposes to use a probability graph model to calculate an Aggregated Prior guided by Multiple Planes, enhancing depth accuracy at boundaries and completeness in textureless objects.

**Strengths:**

- The paper proposes a method that combines the Segment Anything Model (SAM) to assist the PatchMatch process, enhancing the accuracy and completeness of the model by introducing Boundary Plane and Object Plane.
- The proposed method maintains low runtime and memory consumption and is suitable for handling large-scale scenarios.
- The paper includes many experiments and comparisons to evaluate the performance of the proposed method.

**Limitations:**

- As shown in Table 1 and Figure 7 (d), the experimental results of the proposed algorithm do not have obvious performance gain over the existing work like [1], the memory consumption of the proposed method is also higher.
- Although the authors stated in the abstract that they were the first to introduce SAM into PM pipelines, similar approaches have been adopted in previous work [2].

[1] Wang Y, Zeng Z, Guan T, et al. Adaptive patch deformation for textureless-resilient multi-view stereo[C]//Proceedings of the IEEE/CVF Conference on Computer Vision and Pattern Recognition. 2023.

[2] Yuan Z, Cao J, Li Z, et al. SD-MVS: Segmentation-Driven Deformation Multi-View Stereo with Spherical Refinement and EM optimization[C]//Proceedings of the AAAI Conference on Artificial Intelligence. 2024.

**Suitability:**

2

---

### Meta-Review · Area_Chair_vmiT · 2024-06-30

**Recommendation:** Accept (Poster)
**Confidence:** 5

**Metareview:**

This paper originally received contrasting ratings, 2 BA and WR, and after rebuttal they became all positive, 3 BA.
Reviewers criticisms regard the request of clarification of some part of the methodological pipeline and, above all, the experimental analysis, performance, and lack of significant ablations and weal comparative analysis.

After rebuttal, all the raised shortcomings seem reasonably addressed, leading to all positive ratings. For this reason, this paper can be accepted for publication to ACM MM 2024 conference.